# PERTURBATION-RESTRAINED SEQUENTIAL MODEL EDITING

**Jun-Yu Ma**[1,2]**, Hong Wang**[1]**, Hao-Xiang Xu**[1,2]**, Zhen-Hua Ling**[1,2]**, Jia-Chen Gu**[3]*

[1]University of Science and Technology of China
[2]National Engineering Research Center of Speech and Language Information Processing
[3]University of California, Los Angeles

`{mjy1999,wanghong1700,nh2001620}@mail.ustc.edu.cn,`
`zhling@ustc.edu.cn,gujc@ucla.edu`

## ABSTRACT

Model editing is an emerging field that focuses on updating the knowledge embedded within large language models (LLMs) without extensive retraining. However, current model editing methods significantly compromise the general abilities of LLMs as the number of edits increases, and this trade-off poses a substantial challenge to the continual learning of LLMs. In this paper, we first theoretically analyze that the factor affecting the general abilities in sequential model editing lies in the *condition number* of the edited matrix. The condition number of a matrix represents its numerical sensitivity, and therefore can be used to indicate the extent to which the original knowledge associations stored in LLMs are perturbed after editing. Subsequently, statistical findings demonstrate that the value of this factor becomes larger as the number of edits increases, thereby exacerbating the deterioration of general abilities. To this end, a framework termed **P**erturbation **R**estraint on **U**pper bou**N**d for **E**diting (PRUNE) is proposed, which applies the condition number restraints in sequential editing. These restraints can lower the upper bound on perturbation to edited models, thus preserving the general abilities. Systematically, we conduct experiments employing three editing methods on three LLMs across four downstream tasks. The results show that PRUNE can preserve general abilities while maintaining the editing performance effectively in sequential model editing. The code are available at https://github.com/mjy1111/PRUNE.

## 1 INTRODUCTION

Despite the remarkable capabilities of large language models (LLMs), they encounter challenges such as false or outdated knowledge, and the risk of producing toxic content (Zhang et al., 2023; Peng et al., 2023; Ji et al., 2023; Huang et al., 2023). Given the high cost of retraining LLMs to address these issues, there has been a surge in focus on *model editing* (Dai et al., 2022; Meng et al., 2022; Mitchell et al., 2022a;b; Meng et al., 2023; Zhang et al., 2024; Hu et al., 2024; Ma et al., 2024; Jiang et al., 2025; Fang et al., 2025), which aims at updating the knowledge of LLMs cost-effectively. Existing editing methods can be roughly classified into either *parameter-modifying* methods (Mitchell et al., 2022a; Meng et al., 2023) that directly modify a small subset of model parameters, or *parameter-preserving* methods (Mitchell et al., 2022b; Yu et al., 2024) that integrate additional modules without altering the model parameters. In this paper, we study the parameter-modifying editing methods.

Sequential model editing involves making successive edits to the same model over time to continuously update knowledge, as illustrated in Figure 1(a). Recent studies (Gu et al., 2024; Gupta et al., 2024b; Lin et al., 2024; Gupta et al., 2024a) indicate that parameter-modifying editing methods significantly compromise the general abilities of LLMs as the number of edits increases, such as summarization, question answering, and natural language inference. However, these studies neither provide a theoretical analysis of the bottleneck of the general abilities of the edited models, nor propose a solution to preserve these abilities in sequential editing. These affect the scalability of model editing and pose a substantial challenge to the continual learning of LLMs.

---

*Corresponding author.

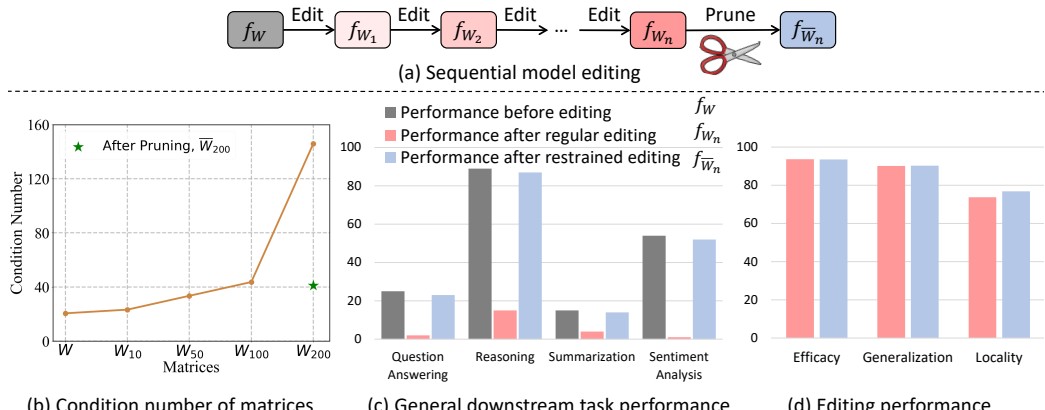

Figure 1: (a) Illustration of sequential model editing. (b) The condition number of edited matrix rapidly increases as the number of edits increases. (c) Comparison of general downstream task performance before editing, after regular editing, and after restrained editing by PRUNE. (d) Comparison of editing performance after regular editing and after restrained editing by PRUNE. $f_W$, $f_{W_n}$ and $f_{\overline{W}_n}$ denote the models that are unedited, regularly edited $n$ times, and restrainedly edited by PRUNE respectively. $W$ is denoted as a matrix to be edited.

In light of the above issues, we first theoretically analyze through matrix perturbation theory (Vaccaro, 1994; Wedin, 1972) to elucidate a crucial factor affecting the general abilities during sequential editing: the *condition number* (Smith, 1967; Dedieu, 1997; Sun, 2000) of the edited matrix. The condition number of a matrix represents its numerical sensitivity and therefore can be used to indicate the extent to which the original knowledge associations stored in LLMs are perturbed after editing. As shown in Figure 1(b), statistical findings demonstrate that the condition number of the edited matrix substantially increases as the number of edits increases, thereby exacerbating the perturbation of original knowledge and the deterioration of general abilities. Therefore, we assume that the bottleneck of the general abilities during sequential editing lies in the escalating value of the condition number.

Towards continual and scalable model editing, we propose **P**erturbation **R**estraint on **U**pper bou**N**d for **E**diting (PRUNE) based on the above analysis, which applies the condition number restraints in sequential editing to preserve general abilities and maintain new editing knowledge simultaneously. Specifically, the condition number of the edited matrix is restrained by reducing the large singular values (Albano et al., 1988; Wall et al., 2003) of the edit update matrix. Consequently, the upper bound on perturbation to the edited matrix is lowered, thus reducing the perturbation to the original knowledge associations and preserving the general abilities of the edited model, as shown in Figure 1(c). Additionally, we observe that these larger singular values often encapsulate redundant editing overfitting information, so regularizing them will not affect the newly editing knowledge, as shown in Figure 1(d). In this way, the new editing knowledge is embedded into LLMs without affecting their original general abilities. Overall, the proposed editing framework requires only minimal computing resources, and is adaptable to be coupled with multiple existing editing methods.

To validate the effectiveness of the proposed PRUNE, our study comprehensively evaluates the edited LLMs for both general abilities and editing performance in sequential editing scenarios. Extensive research involves **three popular editing methods**, including MEND (Mitchell et al., 2022a), ROME (Meng et al., 2022), and MEMIT (Meng et al., 2023), which are analyzed based on **three LLMs** including GPT-2 XL (1.5B) (Radford et al., 2019), LLaMA-2 (7B) (Touvron et al., 2023), and LLaMA-3 (8B). **Four downstream tasks** including reasoning (Cobbe et al., 2021), summarization (Gliwa et al., 2019), open-domain QA (Kwiatkowski et al., 2019), and natural language inference (Dagan et al., 2005) are employed to demonstrate the impact of editing on the general abilities of LLMs. Experimental results demonstrate that the proposed PRUNE can preserve considerable general abilities and maintain almost all editing performance in sequential editing.

In essence, our research offers three contributions: (1) This study theoretically analyzes that the escalating value of the condition number of the edited matrix is the bottleneck of sequential model editing. (2) The PRUNE based on the analysis is proposed to preserve the general abilities of the edited model while retaining the editing knowledge. (3) Experimental results including both editing and downstream task performance across three editing methods on three LLMs demonstrate the effectiveness of PRUNE.

## 2 RELATED WORK

**Model Editing Methods**  From the perspective of whether the model parameters are modified, existing editing methods can be divided into *parameter-modifying* (Mitchell et al., 2022a; Meng et al., 2022; 2023; Dai et al., 2022) and *parameter-preserving* methods (Mitchell et al., 2022b; Hartvigsen et al., 2023; Yu et al., 2024). This paper focuses on the former. Previous works have investigated the role of MLP layers in Transformer, showing that MLP layers store knowledge, which can be located in specific neurons and edited (Geva et al., 2021; Da et al., 2021; Geva et al., 2022). KE (Cao et al., 2021) and MEND (Mitchell et al., 2022a) train a hypernetwork to get gradient changes to update model parameters (Mitchell et al., 2022a). Besides, Meng et al. (2022) and Meng et al. (2023) used Locate-Then-Edit strategy, which first located multi-layer perceptron (MLP) storing factual knowledge, and then edited such knowledge by injecting new key-value pair in the MLP module. Parameter-preserving methods do not modify model weights but store the editing facts with an external memory. For example, Mitchell et al. (2022b) stored edits in a base model and learned to reason over them to adjust its predictions as needed.

**Model Editing Evaluation**  Some works investigate the paradigm for model editing evaluation (Zhong et al., 2023; Cohen et al., 2024; Ma et al., 2023; Li et al.; Hase et al., 2023; Wu et al., 2023; Gandikota et al., 2023; Ma et al., 2024). Cohen et al. (2024) introduced the ripple effects of model editing, suggesting that editing a particular fact implies that many other facts need to be updated. Ma et al. (2023) constructed a new benchmark to assess the edited model bidirectionally. Besides, Li et al. explored two significant areas of concern: Knowledge Conflict and Knowledge Distortion. These early studies mainly evaluate edited models per edit rather than sequentially, and they focus narrowly on basic factual triples. Recently, some works assess the impact of editing methods on the general abilities of LLMs in sequential editing scenarios. These studies (Gu et al., 2024; Gupta et al., 2024b; Lin et al., 2024; Yang et al., 2024; Gupta et al., 2024a;c) have conducted comprehensive experiments, showing the parameter-modifying methods significantly degrade the model performance on downstream tasks.

**Matrix Perturbation Theory**  It plays a crucial role in the field of artificial intelligence (AI) by providing a systematic framework to understand the impact of small changes or perturbations in various AI algorithms and models. Some studies (Harder et al., 2020; Qin et al., 2022; Singh et al., 2024) delve into the interpretability of LLMs, revealing how minor alterations in input features or model parameters influence the model's predictions. This understanding helps uncover significant feature connections within the model architecture. Moreover, it has been instrumental in assessing and enhancing the robustness of models (Chen et al., 2023; 2024). Furthermore, Bird et al. (2020) and Dettmers et al. (2023) have employed it for sensitivity analysis to identify critical factors affecting algorithm performance. It also contributes to the development of efficient optimization techniques (Li et al., 2020; Jiang et al., 2024), improving convergence rates and stability of optimization algorithms.

Compared with previous works (Meng et al., 2022; 2023; Yao et al., 2023; Gu et al., 2024; Gupta et al., 2024b; Lin et al., 2024) that are the most relevant, a main difference should be highlighted. They neither theoretically investigate the reasons for general ability degradation, nor propose effective methods to maintain these abilities during sequential editing. In contrast, our study makes the first attempt to theoretically explore the bottleneck of general abilities in sequential editing and proposes the PRUNE framework to preserve these abilities for continual model editing.

## 3 ANALYSIS ON BOTTLENECK OF SEQUENTIAL MODEL EDITING

### 3.1 PRELIMINARY

**Model Editing**  This task involves modifying the memorized knowledge contained in LMs. Various kinds of complex learned beliefs such as logical, spatial, or numerical knowledge are expected to be edited. In this paper, following previous work (Meng et al., 2022; Zhong et al., 2023; Meng et al., 2023; Zhang et al., 2024), we study editing factual knowledge in the form of (subject $s$, relation $r$, object $o$), e.g., ($s$ = *United States, r = President of, o = Donald Trump*). An LM is expected to recall a memory representing $o$ given a natural language prompt $p(s, r)$ such as "*The President of the United States is*". Editing a fact is to incorporate a new knowledge triple $(s, r, o^*)$ in place of the current one $(s, r, o)$. An edit is represented as $e = (s, r, o, o^*)$ for brevity. Given a set of editing facts

$\mathcal{E} = \{e_1, e_2, \ldots\}$ and an original model $f_{\theta_0}$, sequential model editing operationalizes each edit after the last edit[1], i.e., $K(f_{\theta_{n-1}}, e_n) = f_{\theta_n}$, where $f_{\theta_n}$ denotes the model after $n$ edits.

**Singular Value Decomposition**  SVD (Albano et al., 1988) is a fundamental and effective matrix factorization technique for analyzing matrix structures. Formally, an SVD of a matrix $W \in \mathbb{R}^{p \times q}$ is given by $W = U\Sigma V^{\mathrm{T}}$, where $U = [u_1, u_2, \ldots, u_p] \in \mathbb{R}^{p \times p}$, $V = [v_1, v_2, \ldots, v_q] \in \mathbb{R}^{q \times q}$, and $\Sigma \in \mathbb{R}^{p \times q}$. $u_i$ and $v_i$ are the column vectors of $U$ and $V$, and constitute an orthonormal basis of $\mathbb{R}^p$ and $\mathbb{R}^q$ respectively. $\Sigma$ is a diagonal matrix whose diagonal entries are given by the singular values of $W$ in descending order. Additionally, the SVD of $W$ could also be formulated as: $W = \sum_{i=1}^{\min\{p,q\}} \sigma_i u_i v_i^{\mathrm{T}}$, where $\sigma_i$ is singular value, and $\sigma_1 \geq \sigma_2 \geq \ldots \geq \sigma_{\min\{p,q\}} \geq 0$. In the scenario of this paper, $W$ is a full-rank matrix, so $\sigma_{\min\{p,q\}} > 0$.

## 3.2  Matrix Perturbation Theory Analysis

Previous works (Geva et al., 2021; Meng et al., 2022; Gupta et al., 2023; Wang et al., 2024) have analyzed and located that the MLP modules in Transformer (Vaswani et al., 2017) store various kinds of knowledge (Pearl, 2001; Vig et al., 2020). The MLP module of the $l$-th Transformer layer consists of two projection layers, where the first and second layers are denoted as $W_{fc}^l$ and $W_{proj}^l$ respectively. $W_{proj}^l$ is considered as a linear associative memory which stores knowledge in the form of key-value pairs $(k_i, v_i)$, and is usually regarded as the editing area (Meng et al., 2022; 2023). In this paper, $W_{proj}^l$ is denoted as $W$ for brevity. $W$ is assumed to store many key-value pairs $P = \{(k_i, v_i) \mid i = 1, 2, \ldots\}$ which satisfies $W k_i = v_i$, where $k_i \in \mathbb{R}^q$ and $v_i \in \mathbb{R}^p$. Assuming $|\mathcal{E}| = N$ in sequential model editing, an edit update matrix $\Delta W_j$ is calculated for the edit $e_j$ and added to $W$, which can be formulated as: $W_N = W + \sum_{j=1}^{N} \Delta W_j$ with $\Delta W_j$ calculated from $f_{\theta_{j-1}}$.

**Problem Modeling**  To explore the reasons for the general ability degradation of edited models, we begin by noting that most of the key-value pairs of $P$ correspond to facts unrelated to editing. For the sake of analysis, only the matrix $W$ of a single layer is assumed to be modified. We intuitively hypothesize that for the facts that are irrelevant to the editing fact, the cumulative modifications applied during sequential model editing may lead to significant mismatches in the associations between the original key-value pairs $P$. Specifically, consider a key-value pair $(k_i, v_i) \in P$. After applying an edit $e_j$ that generates $\Delta W_j$ and adding it to $W$, if the extracted value $v_i$ remains unchanged, the corresponding key $k_i$ needs to be adjusted with an adjustment denoted as $\Delta k_i^j$. Mathematically, this can be represented as[2] $W_N(k_i + \sum_{j=1}^{N} \Delta k_i^j) = v_i$ after $N$ edits. However, during the editing process, it's challenging to guarantee such adjustments completely, leading to inaccuracies in the knowledge extracted from the edited model. To delve deeper, let's analyze how the key $k_i$ changes (i.e., $\sum_{j=1}^{N} \Delta k_i^j$) when its corresponding value $v_i$ remains unchanged after $N$ edits.

**Perturbation Analysis of Single Edit**  According to matrix perturbation theory (Luo & Tseng, 1994; Vaccaro, 1994; Wedin, 1972), the edit update matrix $\Delta W$ from an edit can be regarded as a perturbation[3] for $W$, so we first analyze the situation where $W \in \mathbb{R}^{p \times q}$ is appended with a perturbation $\Delta W$. Define $W^{\dagger}$ is the generalized inverse (Stewart & Sun, 1990) of $W$, $\| * \|$ represents 2-norm, and $\tilde{W} = W + \Delta W$.

**Theorem 3.1** Consider $Wk = v$, there exists $\Delta k$ such that $\tilde{k} = k + \Delta k$ satisfies $\tilde{W}\tilde{k} = v$. Let $k = W^{\dagger}v$ and $\tilde{k} = \tilde{W}^{\dagger}v$, and $\Delta W$ is an acute perturbation of $W$. Then:

$$\frac{\|\Delta k\|}{\|k\|} = \frac{\|k - \tilde{k}\|}{\|k\|} \leq \hat{\kappa} \frac{\|\Delta E_{11}\|}{\|W\|} + \Psi_2\left(\frac{\hat{\kappa}\Delta E_{12}}{\|W\|}\right) + \hat{\kappa}^2 \frac{\|\Delta E_{12}\|}{\|W\|}\left(\eta^{-1}g(v) + \frac{\|\Delta E_{21}\|}{\|W\|}\right), \quad (1)$$

where $\Delta E_{11}$, $\Delta E_{12}$, and $\Delta E_{21}$ are directly related to $\Delta W$. $\Psi_2(F)$ is a monotonically increasing function of $\|F\|$ and $g(v)$ is a function about $v$. $\hat{\kappa} = \|W\|\|\tilde{W}_{11}^{-1}\|$, where $\tilde{W}_{11}$ is square and related to the reduced form of $W$. Each term on the right-hand side involves $\hat{\kappa}$, which means that the upper

---

[1]This paper studies editing a single fact at a time and leaves the exploration of batch editing as future work.
[2]As $W_j \in \mathbb{R}^{p \times q}$, and we observed $p < q$ in LLMs, so there will be $\Delta k_i^j$ that satisfies this formula.
[3]We obtained some $\Delta W_j$ and found $\|\Delta W_j\| \ll \|W\|$, which satisfies the definition of perturbation.

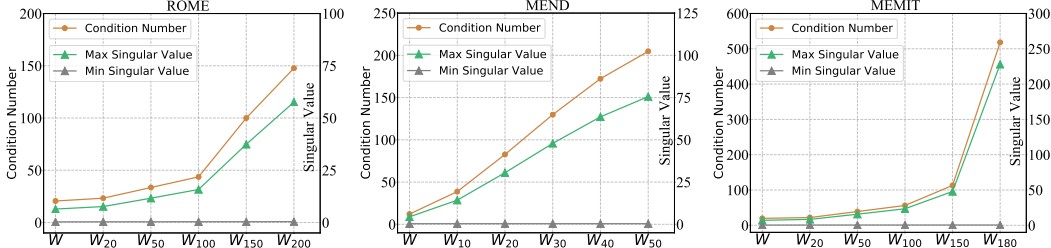

Figure 2: The condition number, maximum singular value and minimum singular value of the edited matrix in sequential editing. Three editing methods including ROME, MEND, and MEMIT are used to edit LLaMA-2 (7B) on the COUNTERFACT (Meng et al., 2022) dataset. For editing methods that modify the parameters of multiple MLP layers, one of them is randomly selected for illustration. $W$ and $W_n$ denote the unedited and edited matrices respectively.

bound on the perturbation of the vector $k$ is constrained by $\hat{\kappa}$. Readers can refer to Appendix A.3 for the details and proof of this theorem. However, calculating $\|\tilde{W}_{11}^{-1}\|$ involves the reduced form of $W$, which incurs unnecessary additional overhead. Therefore, we consider the following theorem and give an alternative estimation.

**Theorem 3.2** Let $\kappa = \|W\|\|W^\dagger\|$, and suppose that $\gamma \equiv 1 - \frac{\kappa\|\Delta E_{11}\|}{\|W\|} > 0$. Then:

$$\|\tilde{W}^\dagger\| \leq \frac{\|W^\dagger\|}{\gamma}. \tag{2}$$

According to Theorem 3.2, $\|\tilde{W}_{11}^{-1}\| \leq \frac{\|W_{11}^{-1}\|}{\gamma} = \frac{\|W^\dagger\|}{\gamma}$, so $\hat{\kappa} \leq \frac{\kappa}{\gamma}$. Here $\kappa = \|W\|\|W^\dagger\| = \frac{\sigma_{max}}{\sigma_{min}}$ is the **condition number** of $W$, where $\sigma_{\max}$ and $\sigma_{\min}$ are the maximum and minimum singular values of $W$, respectively. Combining Theorem 3.1, we know that the larger $\kappa$ is, the greater the upper bound on the perturbation of the vector $k$. Readers can refer to **Appendix A** for the full theoretical analysis.

### 3.3 TREND OF THE CONDITION NUMBER DURING SEQUENTIAL EDITING

As mentioned above, we have analyzed that the condition number of the edited matrix can be used to indicate the upper bound on the perturbation of the key-value pair associations by a single edit. In order to explore the impact of sequential model editing on these associations, the change trend of the condition number of the edited matrix during sequential editing is illustrated in Figure 2.

Surprisingly, we observed that regardless of the editing methods employed, the condition number of the edited matrix exhibited a rapid increase as the number of edits increased, particularly after a large number of edits. According to Theorem 3.1, the adjustment norm $\|\Delta k_i^n\|_2$ corresponding to the $n$-th edit tends to increase as the number of edits $n$ increases. Therefore, we can draw two conclusions: (1) As more edits are performed, the upper bound of the perturbation caused by a new single edit to the key-value pair associations increases. (2) During the sequential model editing process, the cumulative perturbation of these edits will become larger and larger. These factors further disrupt the stored original knowledge and exacerbate the deterioration of general abilities. As the second conclusion is easy to understand, here is an example for the first point. From the first subfigure of Figure 2, we can observe that the condition number of the $W_{200}$ matrix after the 200th edit is significantly higher than that of the unedited matrix $W$. Therefore, the perturbation of the model caused by the 201st edit is likely to be much greater than the perturbation of the model caused by the 1st edit.

## 4 PRUNE: PERTURBATION RESTRAINT ON UPPER BOUND FOR EDITING

**Motivation** According to the analysis in Section 3, the bottleneck of the general abilities during sequential editing lies in the escalating value of the condition number. Assuming a set of edits $\{e_i\}$ and their corresponding edit update matrices $\{\Delta W_i\}$, the information contained in these edit update matrices coordinates with each other to a certain extent since the parametric knowledge of LLMs is distributional rather than independent. This editing overfitting is reflected in SVD, where the largest singular value of the edited matrix $W_N$ becomes significantly large after the addition of these edit

update matrices. To illustrate this, consider an extreme example: suppose we make $N$ edits, where each edit changes the answer to the question "Who is the president of the United States?" to "Biden". Each edit update matrix is denoted as $\Delta W_1$, and its maximum singular value is $\delta_{max}$. Then the sum of the $N$ edit update matrices is $N\Delta W_1$, and its maximum singular value is $N\delta_{max}$, which is amplified by $N$ times. Therefore, our goal is to reduce the editing overfitting in edited matrix $W_N$ as much as possible while also retaining valuable editing information. In this section, a framework termed Perturbation Restraint on Upper bouNd for Editing (PRUNE) is proposed, which applies the condition number restraints to preserve general abilities and maintain new editing knowledge.

**Principle**  Given an edited matrix with $N$ edits, $W_N = W + \sum_{j=1}^{N} \Delta W_j$, as shown in Figure 2, its maximum singular value is constantly increasing, while the minimum singular value is basically unchanged as the number of edits $N$ increases. This directly leads to the increasing condition number of the edited matrix. Therefore, our motivation is to restrain the large singular value of the edited matrix to lower the upper bound on the perturbation. If we directly perform SVD operation on $W_N$ and reduce its singular values, the original $W$ will be inevitably destroyed. Consequently, an analysis of the singular values of $\sum_{j=1}^{N} \Delta W_j$ is conducted, and the results in Table 1

Table 1: The maximum singular values of $\sum_{j=1}^{N} \Delta W_j$ with three edting methods. Other settings are the same as those illustrated in Figure 2.

| Edits ($N$) | ROME | MEMIT | MEND |
|---|---|---|---|
| 10 | 7.25 | 7.46 | 14.08 |
| 50 | 11.38 | 15.63 | 75.53 |
| 100 | 15.62 | 23.39 | 127.89 |
| 200 | 57.61 | 935 | 191.04 |

present that its maximum singular value becomes very large when $N$ is large. Since the singular values of $W$ are relatively small, we can assume that the large maximum singular value of $\sum_{j=1}^{N} \Delta W_j$ is the main reason why the maximum singular value of $W_N$ is large, our method therefore aims to restrain the large singular values of $\sum_{j=1}^{N} \Delta W_j$.

**Design**  Firstly, SVD is operated on the original $W$ and $\sum_{j=1}^{N} \Delta W_j$ respectively as:

$$W = \sum_{i=1}^{\min\{p,q\}} \sigma_i u_i v_i^{\mathrm{T}}, \qquad \sum_{j=1}^{N} \Delta W_j = \sum_{i=1}^{\min\{p,q\}} \hat{\sigma}_i \hat{u}_i \hat{v}_i^{\mathrm{T}}. \tag{3}$$

This paper considers $W$ to be the main part, and any singular value in $\sum_{j=1}^{N} \Delta W_j$ should be ensured not to obviously exceed the maximum singular value of $W$. Subsequently, if any singular value $\hat{\sigma}_i$ of $\sum_{j=1}^{N} \Delta W_j$ is greater than the maximum singular value of $W$, it will be restrained with a function $F$, otherwise it remains unchanged, which could be formulated as:

$$\overline{\sigma}_i = \begin{cases} F(\hat{\sigma}_i), & \text{if } \hat{\sigma}_i > max\{\sigma_i\}, \\ \hat{\sigma}_i, & \text{if } \hat{\sigma}_i \leq max\{\sigma_i\}. \end{cases} \tag{4}$$

$$F(\hat{\sigma}_i) = \log_\alpha(\hat{\sigma}_i) - \log_\alpha(max\{\sigma_i\}) + max\{\sigma_i\}. \tag{5}$$

In the main paper, we use the $\log$ function in $F$ to restrain $\hat{\sigma}_i$. Here $\alpha$ is a hyperparameter to control the degree of restraints, readers can refer to Appendix B.3 for its details for experiments. Besides, we also provide the definition and results of $\text{linear}$ function in Appendix C.3. Finally, we obtain the restrained edited matrix $\overline{W}_N$ to replace $W_N$:

$$\overline{W}_N = W + \sum_{i=1}^{\min\{p,q\}} \overline{\sigma}_i \hat{u}_i \hat{v}_i^{\mathrm{T}}. \tag{6}$$

In this way, the condition number of the edited matrix is reduced (see Appendix C.4) and the upper bound on perturbation is significantly restrained. It is worth noting that the PRUNE proposed here is only used once in Section 5, but can actually be used multiple times in the editing process to better maintain the general ability. We provide a comparison of the two strategies in Appendix C.6.

## 5 EXPERIMENTS

In this section, both the downstream task performance and editing performance of three editing methods on three LLMs were evaluated in sequential model editing. The proposed PRUNE was plug-and-play which can be coupled with these editing methods.

## 5.1 BASE LLMS AND EDITING METHODS

Experiments were conducted on three LLMs including **GPT-2 XL** (1.5B) (Radford et al., 2019), **LLaMA-2** (7B) (Touvron et al., 2023) and **LLaMA-3** (8B)[4]. Three popular editing methods were selected as the baselines including **MEND** (Mitchell et al., 2022a), **ROME** (Meng et al., 2022), and **MEMIT** (Meng et al., 2023). Appendix B.1 shows the details of these editing methods.

## 5.2 EDITING DATASETS AND EVALUATION METRICS

To make a more comprehensive evaluation, we used two types of knowledge for editing: factual knowledge and conceptual knowledge. (1) For factual knowledge, two popular model editing datasets Zero-Shot Relation Extraction (ZsRE) (Levy et al., 2017) and COUNTERFACT (Meng et al., 2022) were adopted in our experiments. These two datasets are QA datasets. A key distinction between COUNTERFACT and ZsRE datasets is that ZsRE contains true facts, while COUNTERFACT contains counterfactual examples where the new target has a lower probability when compared to the original answer (Gupta et al., 2024b). (2) For conceptual knowledge, the ConceptEdit dataset (Wang et al., 2024) was adopted. Due to the limitations of computing resources and pages, most of the experiments in this paper were conducted on factual datasets, with the results presented in Sections 5.4 and 5.5. Meanwhile, Section 5.6 provided some results on conceptual datasets. Readers can refer to Appendix B.2 for examples of each dataset.

To assess the editing performance of editing methods, following previous works (Cao et al., 2021; Mitchell et al., 2022a; Meng et al., 2022; 2023; Ma et al., 2024), three fundamental metrics were employed: efficacy, generalization and locality. Given an original model $f_{\theta_0}$, an edited model $f_{\theta_n}$ with $n$ times sequential editing. Define $\mathbb{1}$ as the indicator function. Each edit $e_i = (s_i, r_i, o_i, o_i^*)$ has an editing prompt $p_i$, paraphrase prompts $\mathcal{P}_i^G$, and locality prompts $\mathcal{P}_i^L$.

**Efficacy** validates whether the edited models could recall the editing fact under editing prompt $p_i$. The assessment is based on Efficacy Score (**ES**) representing as: $\mathbb{E}_i[\mathbb{1}[\arg\max_o P_{f_{\theta_n}}(o \,|\, p_i) = o_i^*]]$.

**Generalization** verifies whether the edited models could recall the editing fact under the paraphrase prompts $\mathcal{P}_i^G$ via Generalization Score (**GS**): $\mathbb{E}_i[\mathbb{E}_{p \in \mathcal{P}_i^G}[\mathbb{1}[\arg\max_o P_{f_{\theta_n}}(o \,|\, p) = o_i^*]]]$.

**Locality** verifies whether the output of the edited models for inputs out of editing scope remains unchanged under the locality prompts $\mathcal{P}_i^L$ via Locality Score (**LS**): $\mathbb{E}_i[\mathbb{E}_{p_l \in \mathcal{P}_i^L}[\mathbb{1}[\arg\max_o P_{f_{\theta_n}}(o \,|\, p_l) = o_l]]]$, where $o_l$ was the original answer of $p_l$.

Different from previous studies that assess the edited models after each individual edit (Gupta et al., 2024b; Yao et al., 2023), this paper evaluated whether the final edited models after completing all edits can still recall all preceding edits, which is more challenging and common in real-world.

## 5.3 DOWNSTREAM TASKS, DATASETS AND METRICS

To explore the side effects of sequential model editing on the general abilities of LLMs, four representative tasks with corresponding datasets were adopted for assessment following previous work (Gu et al., 2024; Gupta et al., 2024b; Lin et al., 2024; Zhang et al., 2024), including:

**Reasoning** on the GSM8K (Cobbe et al., 2021), and the results were measured by solve rate.

**Summarization** on the SAMSum (Gliwa et al., 2019), and the results were measured by the average of ROUGE-1, ROUGE-2 and ROUGE-L following Lin (2004).

**Open-domain QA** on the Natural Question (Kwiatkowski et al., 2019), and the results were measured by exact match (EM) with the reference answer after minor normalization as in Chen et al. (2017) and Lee et al. (2019).

**Natural language inference (NLI)** on the RTE (Dagan et al., 2005), and the results were measured by accuracy of two-way classification.

For each dataset, some examples were randomly sampled for evaluation. Details of prompts for each task were shown in Appendix B.4.

---

[4]https://llama.meta.com/llama3/

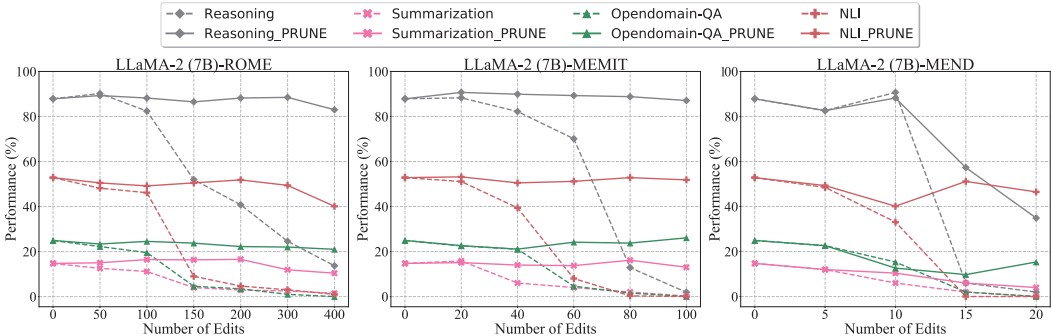

Figure 3: The downstream task performance (%) of models edited by three editing methods with LLaMA-2 (7B) on the ZsRE dataset. The dashed lines refer to the results of the unrestrained editing methods. The solid lines refer to the results of the editing methods coupled with the proposed PRUNE framework. Statistical significance tests were performed to demonstrate that the improvement in PRUNE compared to baseline was statistically significant (t-test with $p$-value <0.05).

## 5.4 GENERAL ABILITIES RESULTS ON FACTUAL KNOWLEDGE

Figure 3 illustrates the downstream task performance of editing methods with LLaMA-2 (7B) on the ZsRE dataset. Due to page limitation, results of other LLMs and factual datasets were put in Appendix C.1. These results were analyzed from the following perspectives.

**Current editing methods significantly compromised general abilities.** As depicted by the dashed lines of Figure 3, both the ROME and MEMIT methods initially maintained relatively stable performance in downstream tasks when the number of edits was small ($\leq 50$). However, as the number of edits surpassed 100, a noticeable decline in performance was observed across all tasks for both methods. Additionally, the MEND method exhibited significant performance degradation after just 20 sequential edits, indicating its inadequacy as a sequential model editing method. Furthermore, when comparing LLMs of different sizes, a general trend emerged: larger models suffered more pronounced compromises in their general abilities when subjected to the same number of edits. For instance, with 300 edits, MEMIT's performance on GPT2-XL remained largely unchanged, whereas it dwindled to nearly 0 on LLaMA-2 and LLaMA-3.

**The performance decline was gradual initially but accelerated with increasing edit count.** This trend aligned with the fluctuation observed in the size of the condition number, as depicted in Figure 2. When the number of edits was small, the condition number was small, and each new edit introduced relatively minor perturbations to the model. However, as the number of edits increased, the condition number underwent a substantial increase. Consequently, each subsequent edit exerted a significant perturbation on the model, leading to a pronounced impairment of its general abilities. These results substantiated the analysis presented in Section 3.3.

**The proposed PRUNE can preserve considerable general abilities.** As shown by the solid lines of Figure 3, when MEMIT was coupled with PRUNE and subjected to 100 edits, its downstream tasks performance remained close to that of the unedited model. However, for the unrestrained MEMIT, downstream task performance had plummeted to nearly 0 by this point. This consistent trend was also observed with ROME and MEND. Nevertheless, for models edited using the unrestrained MEND method, performance degradation was stark after just 10 edits. Even with the addition of PRUNE, preservation could only be extended up to 20 edits. This suggests that while PRUNE effectively preserves general abilities, it does have an upper limit determined by the unrestrained editing method.

## 5.5 EDITING PERFORMANCE RESULTS ON FACTUAL KNOWLEDGE

Figure 4 shows three metrics used for measuring the editing performance with LLaMA-2 (7B) on the ZsRE dataset. Other results were put in Appendix C.2. Three conclusions can be drawn.

**Previous editing facts were forgotten as the number of edits increased.** As shown by the dashed lines of Figure 4, the decline in efficacy and generalization suggests that in sequential editing scenarios, post-edited models gradually forget knowledge acquired from previous edits after a few iterations.

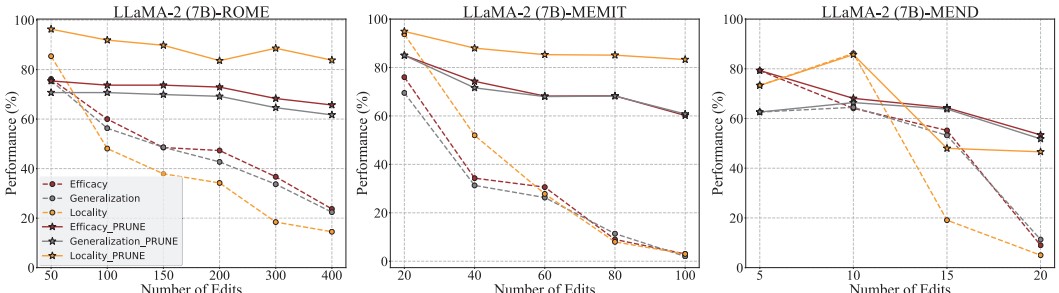

Figure 4: The editing performance (%) of editing methods with LLaMA-2 (7B) on the ZsRE dataset. The dashed lines refer to the results of the unrestrained editing methods. The solid lines refer to the results of the editing methods coupled with the proposed PRUNE. Statistical significance tests were performed to demonstrate that the improvement in PRUNE compared to baseline was statistically significant (t-test with $p$-value <0.05).

Comparing these editing methods, we also observed a notable drop in efficacy and generalization after hundreds of edits with ROME and MEMIT, whereas these values decreased significantly after only 15 edits with MEND. This indicates that in sequential editing scenarios, the MEND method struggled to successfully integrate new knowledge into LLMs after several edits.

**Unrelated facts were perturbed as the number of edits increased.** The locality metric served as an indicator of perturbation for unrelated facts. It became evident that for each editing method, the locality decreased significantly. Additionally, an observation emerged: when the locality of the edited model was low, the performance of downstream tasks was also low. This observation underscores that perturbations of irrelevant knowledge compromise the general abilities of the edited model.

**PRUNE can effectively maintain the editing performance.** This is shown by the solid lines of Figure 4 and could be analyzed from two aspects. On the one hand, when the number of edits was small, the editing performance of each editing method coupled with PRUNE was about the same as the unrestrained method. On the other hand, it significantly mitigated the forgetting of editing facts and the perturbation of irrelevant facts when the number of edits was large during the sequential editing. Specifically, when the number of edits reached 100, the editing performance of MEMIT was very low. But when coupled with PRUNE, its performance remained relatively stable. These observations further validate our motivation in Section 4, demonstrating that the information in the edit update matrices is coordinated, and that performing too many edits can easily result in overfitting. Therefore, applying a certain degree of restraint to edit perturbations can help preserve the model's general abilities while maintaining the editing knowledge.

## 5.6 EDITING WITH CONCEPTUAL KNOWLEDGE

Section 5.4 and 5.5 analyzed the results on factual knowledge. This section conducted some experiments with ROME on conceptual knowledge using the ConceptEdit dataset (Wang et al., 2024) to make a more comprehensive evaluation. For editing performance, in addition to the three basic metrics, this dataset also designed a new metric "Instance Change" to measure whether the instances under the concept changed accordingly when the definition of the concept was changed.

As shown in Table 6, the performance trends of editing and downstream tasks were similar to those observed with the factual datasets. But there are several key differences: (1) When the number of edits was the same, the editing performance of conceptual knowledge was lower than that of factual knowledge. (2) Both editing performance and general abilities deteriorated more quickly than factual knowledge. For example, even if the number of edits was 100, the editing performance and downstream task performance of ROME were very low, while it was still relatively high when editing factual knowledge. (3) The low "Instance Change" indicated that when the definition of a concept was altered, the instances contained in the original concept were still recognized by the model as belonging to that concept. This shows that this editing method primarily modifies the definition without successfully altering the relationship between concepts and instances, which is not reasonable. These findings indicate that conceptual knowledge is more abstract and more difficult to edit than factual knowledge, highlighting the need to explore editing methods for different types of knowledge.

Table 2: Evaluation results (%) of LLaMA-2 (7B) edited by ROME on the ConceptEdit dataset.

| Mode | | General Abilities | | | | Editing Performance | | | |
|---|---|---|---|---|---|---|---|---|---|
| Method | Edits | Reasoning | Summa | Open-QA | NLI | Efficacy | General | Locality | Instance |
| ROME | 20 | 75.13 | 11 | 6.50 | 24.7 | 49.15 | 52.58 | 35.68 | 25 |
| | 50 | 20.67 | 4.90 | 1.50 | 0.7 | 55.42 | 49.45 | 19.94 | 12 |
| | 100 | 12.29 | 4.7 | 0.77 | 0 | 28.25 | 30.18 | 5.68 | 10 |
| | 200 | 0 | 4.62 | 0 | 0 | 10.14 | 8.65 | 5.31 | -8.99 |
| ROME+PRUNE | 20 | 89.38 | 14.34 | 23.37 | 63.54 | 75.66 | 58.35 | 71.7 | 25 |
| | 50 | 85.15 | 14.06 | 25.29 | 50.52 | 56.51 | 45.55 | 73.16 | 8 |
| | 100 | 90.78 | 13.75 | 21.46 | 53.17 | 46.22 | 42.26 | 64.06 | 20 |
| | 200 | 72.9 | 10.55 | 22.22 | 46.15 | 35.82 | 34.95 | 46.65 | 32 |

## 5.7 ANALYSIS ON THE FORGETTING OF EDITING FACTS

Section 3 conducted analysis to elucidate the reasons behind the degradation in general abilities with an increasing number of edits. Subsequent experiments quantitatively demonstrated the effectiveness of PRUNE. Here, we delve into qualitative analysis to explain why editing facts are forgotten and how PRUNE can mitigate this forgetting.

Initially, given a set of editing facts $\mathcal{E} = \{e_1, e_2, \ldots\}$, where $|\mathcal{E}| = 200$. ROME was employed for analysis, and the original matrix was defined as $W$. During sequential editing, ROME computed key-value pairs $(k_j^e, v_j^e)$ of *the last subject token* to generate $\Delta W_j$ for each edit $e_j$ to incorporate new facts, satisfying the equation: $W_j \cdot k_j^e = v_j^e$. However, when evaluating editing performance, the edited model obtained from the last edit was utilized, thus computing values[5]: $W_{200} \cdot k_j^e = \hat{v}_j^e$. After adopting PRUNE to ROME, this equation became $\overline{W}_{200} \cdot k_j^e = \overline{v}_j^e$. We hypothesized that if $\hat{v}_j^e$ was similar to $v_j^e$, the editing fact $e_j$ could be maintained.

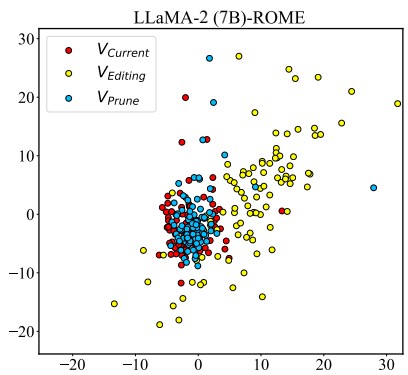

LLaMA-2 (7B)-ROME

Figure 5: 2-dimensional PCA visualization of first 100 values. The model was edited by ROME with LLaMA-2.

Denote $V_{Current} = \{v_j^e\}$, $V_{Editing} = \{\hat{v}_j^e\}$, and $V_{Prune} = \{\overline{v}_j^e\}$. Specifically, these corresponding values of the first 100 edits were used, as they are more prone to be forgotten than the last 100. Principal Component Analysis (PCA) (Gewers et al., 2022) was employed to visualize these values. The first two principal components of each value were calculated and illustrated, as they can represent most of its features (Zheng et al.). As shown in Figure 5, on the one hand, the discrepancy between the principal components of $V_{Current}$ and $V_{Editing}$ was markedly large. This indicates that after 200 edits to the model, the values corresponding to the first 100 facts stored in the edited matrix are severely corrupted, leading to significant forgetfulness. On the other hand, after adopting PRUNE, the discrepancy between the principal components of $V_{Current}$ and $V_{Prune}$ was small. This demonstrates that PRUNE effectively maintains the values and mitigates the forgetting of editing facts.

## 6 CONCLUSION AND LIMITATION

In this paper, a theoretical analysis is firstly conducted to elucidate that the bottleneck of the general abilities during sequential editing lies in the escalating value of the condition number. Subsequently, a plug-and-play framework called PRUNE is proposed to apply restraints to preserve general abilities and maintain new editing knowledge simultaneously. Comprehensive experiments on various editing methods and LLMs demonstrate the effectiveness of this method. We aspire that our analysis and method will catalyze future research on continual model editing.

**Limitation** Firstly, this paper focuses on editing a single fact at a time in sequential model editing, but some works study updating hundreds of facts simultaneously in batch editing. Therefore, investigating batch-sequential editing could enhance the scalability of model editing. Secondly, it is necessary to explore the performance of larger-size models and more editing methods on more downstream tasks.

---

[5]Since ROME only modifies one matrix, the $k_j^e$ remains the same across these edited models.

ACKNOWLEDGMENTS

We would like to express gratitude to the anonymous reviewers for kind comments. This work is funded by the National Science and Technology Major Project (No. 2023ZD0121103).

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

APPENDIX

# A THEORETICAL ANALYSIS BASED ON PERTURBATION THEORY

Here, we provide a detailed analysis and proof of Section 3.2. We begin by introducing some definitions and then present several preliminary lemmas and theorems. These lemmas and theorems are finally used to prove Theorem 3, which is most relevant to our problem discussed in Section 3.2.

## A.1 DEFINITION

We discuss the problem $Ax = b$, where $\tilde{A}$ is a perturbation of $A$ given by $\tilde{A} = A + E$. We assume $b$ remains unchanged and $\tilde{x}$ represents the corresponding change, satisfying $\tilde{A}\tilde{x} = b$. Here $A \in \mathbb{C}^{m \times n}$, $b \in \mathbb{C}^m$.

It is noteworthy that in the following derivation, $A^H$ denotes the conjugate transpose of $A$, $A^\dagger$ represents the generalized inverse of $A$, and $\| * \|$ represents 2-norm (Stewart & Sun, 1990).

To simplify the problem, we apply a rotation. Specifically, let $V = (V_1 \ V_2)$ be a unitary matrix with $R(V_1) = R(A^H)$, and let $U = (U_1 \ U_2)$ be a unitary matrix with $R(U_1) = R(A)$, where $R$ refers to the rank. Then

$$U^H A V = \begin{pmatrix} U_1^H A V_1 & U_1^H A V_2 \\ U_2^H A V_1 & U_2^H A V_2 \end{pmatrix} = \begin{pmatrix} A_{11} & 0 \\ 0 & 0 \end{pmatrix}, \tag{7}$$

where $A_{11}$ is square and nonsingular. If we set

$$U^H E V = \begin{pmatrix} U_1^H E V_1 & U_1^H E V_2 \\ U_2^H E V_1 & U_2^H E V_2 \end{pmatrix} = \begin{pmatrix} E_{11} & E_{12} \\ E_{21} & E_{22} \end{pmatrix}, \tag{8}$$

then

$$U^H \tilde{A} V = \begin{pmatrix} A_{11} + E_{11} & E_{12} \\ E_{21} & E_{22} \end{pmatrix} = \begin{pmatrix} \tilde{A}_{11} & E_{12} \\ E_{21} & E_{22} \end{pmatrix}. \tag{9}$$

We will call these transformed, partitioned matrices the **reduced form** of the problem. Many statements about the original problem have revealing analogues in the reduced form.

In this form, $x$ is replaced by $V^H x$ and $b$ is replaced by $U^H b$. If $x$ and $b$ are partitioned in the forms

$$x = \begin{pmatrix} x_1 \\ x_2 \end{pmatrix}, \quad b = \begin{pmatrix} b_1 \\ b_2 \end{pmatrix}, \tag{10}$$

where $x_1, b_1 \in \mathbb{C}^r$, then

$$x_1 = A_{11}^{-1} b_1 \tag{11}$$

and

$$x_2 = 0. \tag{12}$$

Moreover, the norm of the residual vector

$$r = b - Ax \tag{13}$$

is given by

$$\|r\| = \|b_2\|. \tag{14}$$

Here, we define the symbol $\eta$:

$$\eta = \frac{\|A\|\|x\|}{\|b\|}, \tag{15}$$

and for any $F \in \mathbb{C}^{k \times r}$ ($k \geq r$) the symbol $\Psi(F)$, for the spectral norm:

$$\Psi_2(F) = \frac{\|F\|}{(1 + \|F\|^2)^{1/2}}. \tag{16}$$

## A.2 PRELIMINARY LEMMAS & THEOREMS

After introducing some definitions, we give some preliminary lemmas and theorems, which are used to prove Theorem 3.

**Lemma 1** Let

$$\kappa(A) = \|A\|\|A^{-1}\|$$

be the condition number of $A$. If $\tilde{A}$ is nonsingular, then

$$\frac{\|\tilde{A}^{-1} - A^{-1}\|}{\|\tilde{A}^{-1}\|} \leq \kappa(A)\frac{\|E\|}{\|A\|}. \tag{17}$$

If in addition

$$\frac{\|E\|}{\|A\|}\kappa(A) < 1,$$

then $\tilde{A}$ is perforce nonsingular and

$$\|\tilde{A}^{-1}\| \leq \frac{\|A^{-1}\|}{1 - \kappa(A)\frac{\|E\|}{\|A\|}}. \tag{18}$$

Moreover

$$\frac{\|\tilde{A}^{-1} - A^{-1}\|}{\|A^{-1}\|} \leq \frac{\kappa(A)\frac{\|E\|}{\|A\|}}{1 - \kappa(A)\frac{\|E\|}{\|A\|}}. \tag{19}$$

**Lemma 2** In the reduced form the matrices $A$ and $\tilde{A}$ are acute if and only if $A_{11}$ is nonsingular and

$$E_{22} = E_{21}\tilde{A}_{11}^{-1}E_{12}. \tag{20}$$

In this case, if we set

$$F_{21} = E_{21}\tilde{A}_{11}^{-1} \quad \text{and} \quad F_{12} = \tilde{A}_{11}^{-1}E_{12},$$

then

$$\tilde{A} = \begin{pmatrix} I \\ F_{21} \end{pmatrix} \tilde{A}_{11} \begin{pmatrix} I & F_{12} \end{pmatrix}$$

and

$$\tilde{A}^{\dagger} = \begin{pmatrix} I & F_{12} \end{pmatrix}^{\dagger} \tilde{A}_{11}^{-1} \begin{pmatrix} I \\ F_{21} \end{pmatrix}^{\dagger}. \tag{21}$$

**Lemma 3** The matrix

$$\begin{pmatrix} I \\ F \end{pmatrix}$$

satisfies

$$\left\| \begin{pmatrix} I \\ F \end{pmatrix}^{\dagger} \right\| \leq 1 \tag{22}$$

and

$$\left\| \begin{pmatrix} I \\ F \end{pmatrix}^{\dagger} - \begin{pmatrix} I & 0 \end{pmatrix} \right\| = \Psi_2(F). \tag{23}$$

**Theorem 1** Let $\tilde{A}$ be an acute perturbation of $A$, and let

$$\hat{\kappa} = \|A\|\|\tilde{A}_{11}^{-1}\|. \tag{24}$$

Then

$$\frac{\|\tilde{A}^{\dagger} - A^{\dagger}\|}{\|A^{\dagger}\|} \leq \hat{\kappa}\frac{\|E_{11}\|}{\|A\|} + \Psi_2\left(\frac{\hat{\kappa}E_{12}}{\|A\|}\right) + \Psi_2\left(\frac{\hat{\kappa}E_{21}}{\|A\|}\right). \tag{25}$$

*Proof.* Let

$$I_{21} = \begin{pmatrix} I \\ 0 \end{pmatrix}, \quad I_{12} = (I \quad 0), \tag{26}$$

$$J_{21} = \begin{pmatrix} I \\ F_{21} \end{pmatrix}, \quad J_{12} = (I \quad F_{12}). \tag{27}$$

$\tilde{A}^\dagger = J_{12}^\dagger A_{11}^{-1} I_{21}^\dagger$, hence

$$\tilde{A}^\dagger - A^\dagger = (J_{12}^\dagger - I_{12}^\dagger) A_{11}^{-1} I_{21}^\dagger + J_{12}^\dagger A_{11}^{-1}(J_{21}^\dagger - I_{21}^\dagger) + J_{12}^\dagger(\tilde{A}_{11}^{-1} - A_{11}^{-1}) J_{21}^\dagger. \tag{28}$$

From Lemma 1 we have the following bound:

$$\|J_{12}^\dagger(\tilde{A}_{11}^{-1} - A_{11}^{-1}) J_{21}^\dagger\| \le \|A_{11}^{-1}\|\hat{\kappa}\frac{\|E_{11}\|}{\|A_{11}\|}. \tag{29}$$

By Lemma 3

$$\|(J_{12}^\dagger - I_{12}^\dagger) A_{11}^{-1} I_{21}^\dagger\| \le \|A_{11}^{-1}\|\|J_{12}^\dagger - I_{12}^\dagger\| = \|A_{11}^{-1}\|\Psi_2(F_{12}) \tag{30}$$

$$= \|A_{11}^{-1}\|\Psi_2(\tilde{A}_{11}^{-1} E_{12}) \tag{31}$$

$$\le \|A_{11}^{-1}\|\Psi_2\left(\frac{\hat{\kappa} E_{12}}{\|A\|}\right), \tag{32}$$

and likewise

$$\|J_{12}^\dagger A_{11}^{-1}(J_{21}^\dagger - I_{21}^\dagger)\| \le \|A_{11}^{-1}\|\Psi_2\left(\frac{\hat{\kappa} E_{21}}{\|A\|}\right) \le \|A^\dagger\|\Psi_2\left(\frac{\hat{\kappa} E_{21}}{\|A\|}\right). \tag{33}$$

$\square$

**Theorem 2** In Theorem 1, let

$$\kappa = \|A\|\|A^\dagger\|, \tag{34}$$

and suppose that

$$\|A^\dagger\|\|E_{11}\| < 1, \tag{35}$$

so that

$$\gamma \equiv 1 - \frac{\kappa\|E_{11}\|}{\|A\|} > 0. \tag{36}$$

Then

$$\|\tilde{A}^\dagger\| \le \frac{\|A^\dagger\|}{\gamma}, \tag{37}$$

and

$$\frac{\|\tilde{A}^\dagger - A^\dagger\|}{\|A^\dagger\|} \le \frac{\kappa\|E_{11}\|}{\gamma\|A\|} + \Psi_2\left(\frac{\kappa E_{21}}{\gamma\|A\|}\right) + \Psi_2\left(\frac{\kappa E_{12}}{\gamma\|A\|}\right). \tag{38}$$

*Proof.* From the equation $\tilde{A}^\dagger = J_{12}^\dagger \tilde{A}_{11}^{-1} J_{21}^\dagger$, we have

$$\|\tilde{A}^\dagger\| \le \|J_{12}^\dagger\|\|\tilde{A}_{11}^{-1}\|\|J_{21}^\dagger\| \le \|\tilde{A}_{11}^{-1}\|. \tag{39}$$

By Lemma 1,

$$\|\tilde{A}_{11}^{-1}\| \le \frac{\|A_{11}^{-1}\|}{\gamma} = \frac{\|A^\dagger\|}{\gamma}, \tag{40}$$

which establishes equation 37. Also $\hat{\kappa} \le \frac{\kappa}{\gamma}$, and the inequality equation 38 follows from equation 25.

$\square$

A.3    CORE THEOREM

Finally, we give the core theorem used in main paper. Some symbols and definitions have been claimed in Appendix A.1 and A.2.

**Theorem 3** Let $x = A^\dagger b$ and $\tilde{x} = \tilde{A}^\dagger b$, where $\tilde{A} = A + E$, and $E$ is an acute perturbation of $A$. Then

$$\frac{\|x - \tilde{x}\|}{\|x\|} \leq \hat{\kappa}\frac{\|E_{11}\|}{\|A\|} + \Psi_2\left(\frac{\hat{\kappa}E_{12}}{\|A\|}\right) + \hat{\kappa}^2\frac{\|E_{12}\|}{\|A\|}\left(\eta^{-1}\frac{\|b_2\|}{\|b_1\|} + \frac{\|E_{21}\|}{\|A\|}\right). \quad (41)$$

*Proof.* By Lemma 2, write

$$\tilde{x} - x = J_{12}^\dagger(\tilde{A}_{11}^{-1} - A_{11}^{-1})b_1 + (J_{12}^\dagger - I_{12}^\dagger)A_{11}^{-1}b_1 + J_{12}^\dagger\tilde{A}_{11}^{-1}(J_{21}^\dagger - I_{21}^\dagger)b. \quad (42)$$

Then

$$\|J_{12}^\dagger(\tilde{A}_{11}^{-1} - A_{11}^{-1})b_1\| \leq \hat{\kappa}\frac{\|E_{11}\|}{\|A\|}\|x\|, \quad (43)$$

and

$$\|(J_{12}^\dagger - I_{12}^\dagger)A_{11}^{-1}b_1\| \leq \Psi_2\left(\frac{\hat{\kappa}E_{12}}{\|A\|}\right)\|x\|. \quad (44)$$

Now

$$J_{12}^\dagger\tilde{A}_{11}^{-1}(J_{21}^\dagger - I_{21}^\dagger)b = J_{12}^\dagger\tilde{A}_{11}^{-1}((I + F_{21}^H F_{21})^{-1} - I)b_1 + J_{12}^\dagger\tilde{A}_{11}^{-1}(I + F_{21}^H F_{21})^{-1}F_{21}^H b_2. \quad (45)$$

To bound the first term in equation 45, note that

$$(I + F_{21}^H F_{21})^{-1} - I = -(I + F_{21}^H F_{21})^{-1}F_{21}^H F_{21}.$$

Hence

$$\begin{aligned}
\|J_{12}\tilde{A}_{11}^{-1}((I + F_{21}^H F_{21}) - I)b_1\| &\leq \|\tilde{A}_{11}^{-1}\|\|(I + F_{21}^H F_{21})^{-1}\|\|F_{21}^H\|\|F_{21}b_1\| \\
&\leq \|\tilde{A}_{11}^{-1}\|\|E_{21}\tilde{A}_{11}^{-1}b_1\| \\
&\leq \|\tilde{A}_{11}^{-1}\|\|E_{21}\|^2\|x\| \\
&= \left(\frac{\hat{\kappa}\|E_{21}\|_2}{\|A\|}\right)^2\|x\|.
\end{aligned} \quad (46)$$

For the second term in equation 45 we have

$$\begin{aligned}
\|J_{21}^\dagger\tilde{A}_{11}^{-1}(I + F_{21}^H F_{21})^{-1}F_{21}b_2\| &\leq \|\tilde{A}_{11}^{-1}\|^2\|E_{21}\|\|b_2\| \\
&= \|\tilde{A}_{11}^{-1}\|^2\|E_{21}\|\frac{\|b_2\|}{\|b_1\|}\eta^{-1}\|x\|\|A\| \\
&\leq \eta^{-1}\hat{\kappa}^2\frac{\|E_{21}\|\|b_2\|}{\|A\|\|b_1\|}\|x\|.
\end{aligned} \quad (47)$$

The bound equation 41 follows on combining equation 42–equation 47.

$\square$

Readers can refer to this work (Stewart & Sun, 1990) for more details of perturbation analysis.

Returning to our problem, consider $Wk = v$, where $(k, v) \in P$. Let $\tilde{W} = W + \Delta W$, where $\Delta W$ is the corresponding perturbation matrix. Assuming $v$ remains constant, there exists $\Delta k$ such that $\tilde{k} = k + \Delta k$ satisfies $\tilde{W}\tilde{k} = v$. And we have $k = W^\dagger v$ and $\tilde{k} = \tilde{W}^\dagger v$. Applying Theorem 3, we obtain

$$\frac{\|\Delta k\|}{\|k\|} = \frac{\|k - \tilde{k}\|}{\|k\|} \leq \hat{\kappa}\frac{\|\Delta E_{11}\|}{\|W\|} + \Psi_2\left(\frac{\hat{\kappa}\Delta E_{12}}{\|W\|}\right) + \hat{\kappa}^2\frac{\|\Delta E_{12}\|}{\|W\|}\left(\eta^{-1}\frac{\|v_2\|}{\|v_1\|} + \frac{\|\Delta E_{21}\|}{\|W\|}\right), \quad (48)$$

where $\Delta E_{11}$, $\Delta E_{12}$, $\Delta E_{21}$, and $\Delta W$ are directly related, and each term on the right-hand side involves $\hat{\kappa}$. This means that the relative perturbation of the vector $k$ is constrained by $\hat{\kappa}$. According to Theorem 2, $\hat{\kappa} \leq \frac{\kappa}{\gamma}$, where $\kappa = \|W\|\|W^\dagger\|$ is the condition number of $W$. This indicates that $\kappa$ is a robust indicator of the impact of $\Delta W$ on the vector $k$.

# B EXPERIMENTAL SETUP

## B.1 BASELINE EDITING METHODS

Three popular model editing methods were selected as baselines including:

- **MEND** (Mitchell et al., 2022a)[6]: it learned a hypernetwork to produce weight updates by decomposing the fine-tuning gradients into rank-1 form.
- **ROME** (Meng et al., 2022)[7]: it first localized the factual knowledge at a specific layer in the transformer MLP modules, and then updated the knowledge by directly writing new key-value pairs in the MLP module.
- **MEMIT** (Meng et al., 2023)[8]: it extended ROME to edit a large set of facts and updated a set of MLP layers to update knowledge.

The ability of these methods were assessed based on EasyEdit[9] (Wang et al., 2023), an easy-to-use knowledge editing framework which integrates the released codes and hyperparameters from previous methods.

## B.2 EDITING DATASETS AND EVALUATION METRICS

Table 3 shows the examples of two factual datasets (ZsRE) (Levy et al., 2017) and COUNTER-FACT (Meng et al., 2022). Figure 6 shows an example of ConceptEdit dataset, which is cited from Wang et al. (2024). More details can refer to the original paper of these datasets.

Table 3: The editing datasets of both ZsRE and COUNTERFACT.

| Datasets | Editing prompt |
|---|---|
| ZsRE | Which was the record label for New Faces, New Sounds? |
| COUNTERFACT | In America, the official language is |

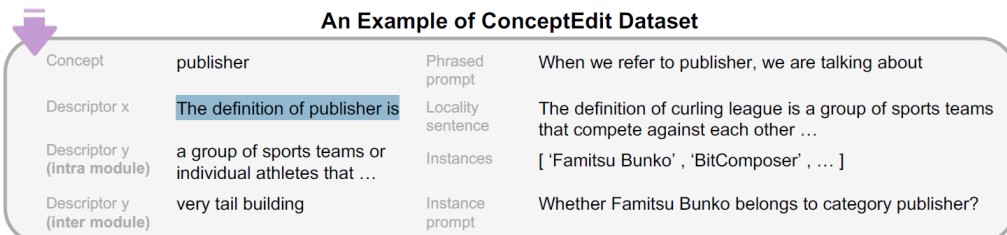

Figure 6: An example of ConceptEdit dataset

Besides, following previous works (Meng et al., 2022; Mitchell et al., 2022a; Meng et al., 2023), the editing performance metrics for the ZsRE and COUNTERFACT datasets are efficacy, generalization and locality, but there are some computational differences. In the main paper, the metrics of editing performance are used for the ZsRE dataset.

For the COUNTERFACT dataset, here are the details:

**Efficacy** validates whether the edited models could recall the editing fact under editing prompt $p_i$. The assessment is based on Efficacy Score (**ES**) representing as: $\mathbb{E}_i[\mathbb{1}[\,P_{f_{\theta_n}}(o_i^* \,|\, p_i) > P_{f_{\theta_n}}(o_i \,|\, p_i)\,]\,]$, where $\mathbb{1}$ is the indicator function.

---

[6]https://github.com/eric-mitchell/mend
[7]https://github.com/kmeng01/rome
[8]https://github.com/kmeng01/memit
[9]https://github.com/zjunlp/EasyEdit

**Generalization** verifies whether the edited models could recall the editing fact under the paraphrase prompts $\mathcal{P}_i^G$ via Generalization Score (**GS**): $\mathbb{E}_i\left[\mathbb{E}_{p\in\mathcal{P}_i^G}[\mathbb{1}[P_{f_{\theta_n}}(o_i^*\,|\,p) > P_{f_{\theta_n}}(o_i\,|\,p)]]\right]$.

**Locality** verifies whether the output of the edited models for inputs out of editing scope remains unchanged under the locality prompts $\mathcal{P}_i^L$ via Locality Score (**LS**): $\mathbb{E}_i\left[\mathbb{E}_{p_l\in\mathcal{P}_i^L}[\mathbb{1}[P_{f_{\theta_n}}(o_l\,|\,p_l) > P_{f_{\theta_n}}(o_i^*\,|\,p_l)]]\right]$, where $o_l$ was the original answer of $p_l$.

### B.3 HYPERPARAMETERS OF PRUNE

When conducting experiments, for different editing methods, LLMs and editing datasets, the hyperparameter $\alpha$ in function $F$ of PRUNE is different. Table 4 shows the details of this hyperparameter. $e$ is the base of the natural logarithm.

Table 4: The hyperparameters $\alpha$ for PRUNE.

| Datasets | Models | ROME | MEMIT | MEND |
|---|---|---|---|---|
| | GPT-2 XL | 1.2 | 1.2 | 1.2 |
| COUNTERFACT | LLaMA-2 | 1.2 | $e$ | 1.2 |
| | LLaMA-3 | 1.5 | $e$ | - |
| ZSRE | LLaMA-2 | 1.2 | $e$ | $e$ |

### B.4 TASK PROMPTS

The prompts for each downstream task were illustrated in Table 5.

Table 5: The prompts to LLMs for evaluating their zero-shot performance on these general tasks.

| |
|---|
| Reasoning: 
 Q: {QUESTION} A: Let's think step by step. {HINT} Therefore, the answer (arabic numerals) is: |
| NLI: 
 {SENTENCE1} entails the {SENTENCE2}. True or False? answer: |
| Open-domain QA: 
 Refer to the passage below and answer the following question. Passage: {DOCUMENT} Question: {QUESTION} |
| Summarization: 
 {DIALOGUE} TL;DR: |

### B.5 EXPERIMENTS COMPUTE RESOURCES

We used NVIDIA A800 80GB GPU for experiments. For LLaMA-2 (7B) and LLaMA-3 (8B), it occupies about 40+GB memory and costs about 3 hours for each editing method to run 200 edits and then to test downstream tasks . For GPT-2 XL (1.5B), it needs 10+GB and costs about 1.5 hours for each editing method to run 200 edits and then to test downstream tasks.

## C EXPERIMENTAL RESULTS

### C.1 RESULTS OF GENERAL ABILITIES

Figure 7, 8 and 9 show the downstream task performance of edited models with GPT-2 XL, LLaMA-2 (7B) and LLaMA-3 (8B) on COUNTERFACT dataset. Due to limitations of computing resources, experiments were conducted using only LLaMA-2 (7B) on the ZSRE dataset. We will supplement experiments with other LLMs in the future.

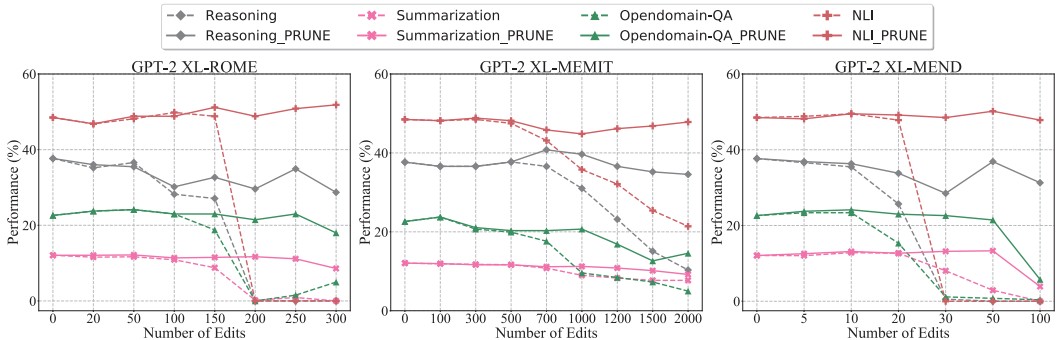

Figure 7: The downstream task performance (%) of models edited by three editing methods with GPT-2 XL on the COUNTERFACT dataset.

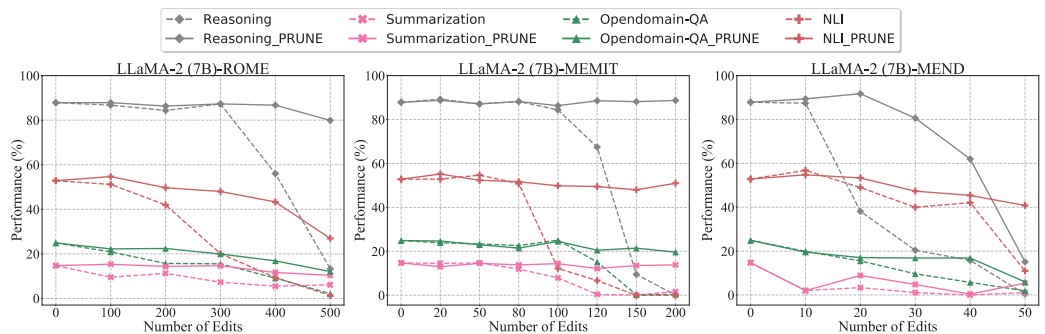

Figure 8: The downstream task performance (%) of models edited by three editing methods with LLaMA-2 (7B) on the COUNTERFACT dataset.

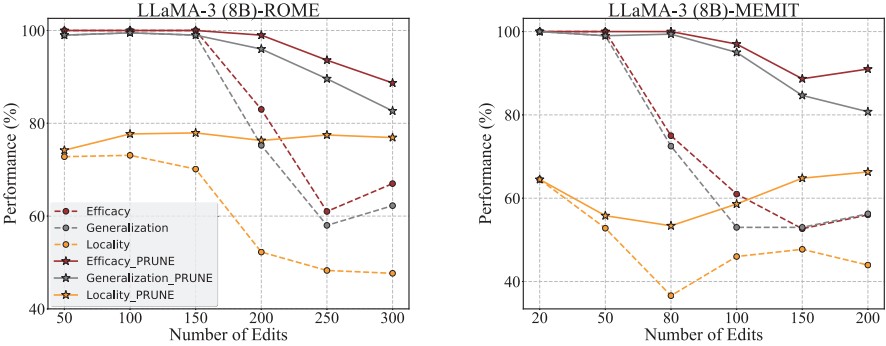

Figure 9: The downstream task performance (%) of models edited by two editing methods with LLaMA-3 (8B) on the COUNTERFACT dataset. Since the code framework EasyEdit used in this paper does not currently support MEND editing on LLaMA-3, there are no results of MEND here.

## C.2 RESULTS OF EDITING PERFORMANCE

Figure 10, 11 and 12 shows the editing performance of edited models with GPT-2 XL, LLaMA-2 (7B) and LLaMA-3 (8B) on COUNTERFACT dataset.

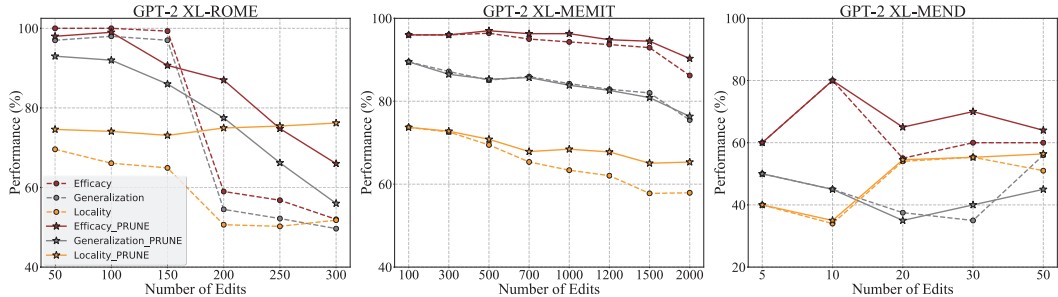

Figure 10: The editing performance (%) of three editing methods with GPT-2 XL on COUNTERFACT.

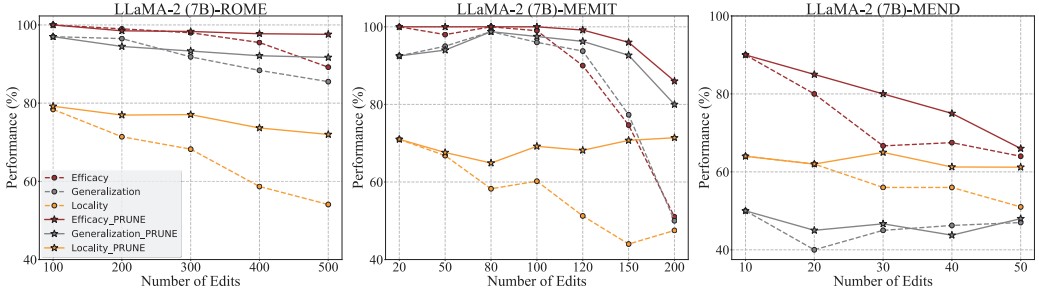

Figure 11: The editing performance (%) of three editing methods with LLaMA-2 (7B) on the COUNTERFACT dataset.

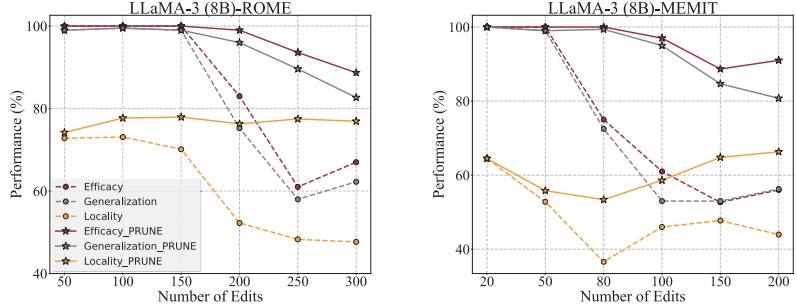

Figure 12: The editing performance (%) of three editing methods with LLaMA-3 (8B) on COUNTERFACT dataset.

## C.3 RESULTS OF ANOTHER FUNCTION FOR PRUNE

In the main paper, $\log$ function is used in $F$ in PRUNE to restrain $\hat{\sigma}_i$. Here we use the linear function, which could be represented as: $F(\hat{\sigma}_i) = \frac{1}{\beta} * \hat{\sigma}_i + \frac{\beta-1}{\beta} * max\{\sigma_i\}$. Here $\beta > 1$ was a hyperparameter and was set as 2 in this section. Figure 13 and 14 respectively show some downstream task performance and editing performance with linear function on COUNTERFACT dataset.

Compared with Figure 7 and 10, we observed that although the linear function in PRUNE played a role in preserving general abilities and maintaining editing performance, its effectiveness was noticeably inferior to that of the $\log$ function when the number of edits was large.

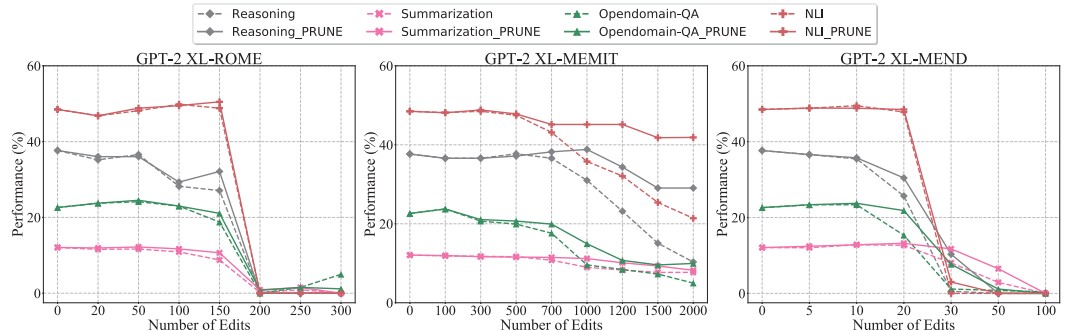

Figure 13: The downstream task performance (%) of models edited by three editing methods with GPT-2 XL on the COUNTERFACT dataset. Here the linear function was used in PRUNE.

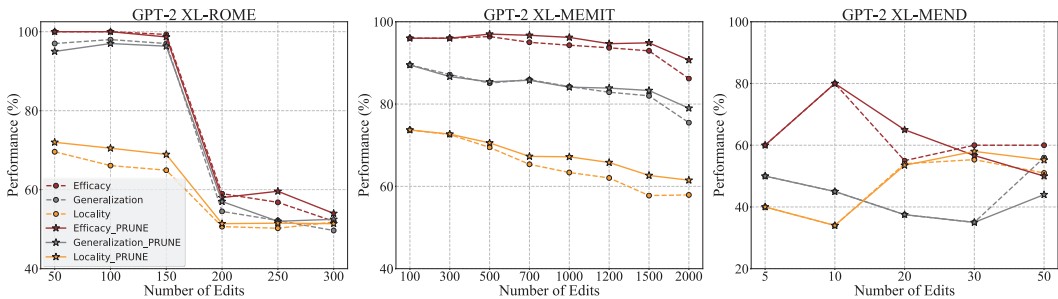

Figure 14: The editing performance (%) of editing methods with GPT-2 XL on the COUNTERFACT dataset. Here the linear function was used in PRUNE.

## C.4 CONDITION NUMBER WITH PRUNE

Figure 15 shows after coupling with PRUNE, the condition number of MEMIT is significantly restrained.

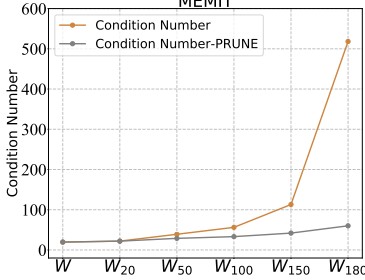

Figure 15: The condition number of MEMIT with LLaMA-2 (7B) on the COUNTERFACT dataset. "-PRUNE" refers to the condition number of MEMIT coupled with the proposed PRUNE.

## C.5 THE CORRELATION BETWEEN CONDITION NUMBER AND GENERAL ABILITIES

Figure 16 simultaneously shows the condition number and general abilities of three editing methods without PRUNE in the sequential editing process. From these experiments, we observed that a dramatic increase in the condition number is often accompanied by a rapid decline in general abilities.

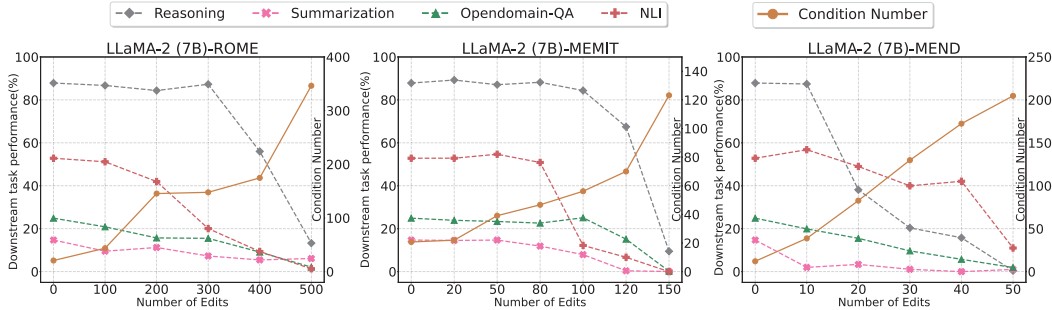

Figure 16: The condition number and downstream task performance of three editing methods with LLaMA-2 (7B) on the COUNTERFACT dataset. Since MEMIT and MEND have multiple parameters to be edited, we randomly selected one of them to calculate the condition number.

## C.6 ENHANCED PRUNE

We find applying multiple operation of PRUNE for longer-term performances will perform better than only once. The results of applying PRUNE once have been shown in our paper. Here, we make a comparison. For convenience, we list some results of using ROME to edit GPT 2-XL on the CounterFact dataset.

Table 6: comparison of applying once operation of PRUNE and applying multiple multiple operation.

| Mode | | General Abilities | | | | Editing Performance | | |
|---|---|---|---|---|---|---|---|---|
| Method | Edits | Reasoning | Summa | Open-QA | NLI | Efficacy | General | Locality |
| ROME+PRUNE (once) | 100 | 30.12 | 12.12 | 22.63 | 49.83 | 99 | 92 | 74.10 |
| | 500 | 27.26 | 10.78 | 17.86 | 45.34 | 56.80 | 46.05 | 72.32 |
| ROME+PRUNE (multiple) | 100 | 30.17 | 11.38 | 22.99 | 48.83 | 100 | 98 | 76.1 |
| | 500 | 32.68 | 11.52 | 22.99 | 51.17 | 84.6 | 78.7 | 75.1 |

## D BROADER IMPACTS

This work offers significant advancements in the field of model editing for LLMs. By addressing the challenge of preserving general abilities while performing sequential edits, PRUNE facilitates continual learning and adaptability in LLMs. This can lead to several positive impacts, such as:

**Enhanced Adaptability.** It enables LLMs to update their knowledge base quickly and accurately without extensive retraining. This adaptability is crucial in dynamic environments where up-to-date information is vital, such as real-time translation services, personalized learning systems, and interactive virtual assistants.

**Resource Efficiency.** By mitigating the need for full retraining, PRUNE significantly reduces computational resources and energy consumption. This aligns with sustainable AI and makes it more feasible to deploy LLMs in resource-constrained settings.

**Improved Performance in Specialized Tasks.** PRUNE's ability to perform targeted edits without compromising overall model performance can enhance LLMs' effectiveness in specialized domains, such as medical diagnostics, legal analysis, and technical support, where precise and updated knowledge is essential.

While this work offers many benefits, there are potential negative societal impacts that must be considered:

**Misuse for Malicious Purposes.** The capability to edit LLMs efficiently could be exploited to inject harmful or biased information into models, thereby spreading disinformation or propaganda. This risk is particularly concerning in applications involving social media and news dissemination where LLMs might generate or amplify misleading content.

**Fairness.** Unintended biases could be introduced during the editing process, potentially exacerbating existing biases in LLMs. This could lead to unfair treatment or misrepresentation of specific

groups, especially if the editing is not conducted with proper oversight and consideration of ethical implications.

**Privacy Concerns.** The ability to update models quickly might also pose privacy risks, as models could be edited to include sensitive or personal information. Ensuring that editing processes do not compromise individual privacy is critical, particularly in applications involving personal data.

To mitigate these potential negative impacts, several strategies could be implemented:

**Gated Release and Monitoring.** Limiting access to the framework through gated releases and monitoring its usage can help prevent misuse.

**Bias and Fairness Audits.** Conducting regular audits to assess and address biases in the model editing process can help ensure that edits do not unfairly impact any specific group. Developing guidelines for ethical editing practices is also essential.

**Privacy Protection Measures.** Establishing clear protocols for handling sensitive data during the editing process can help protect privacy. Anonymization and encryption techniques should be employed to safeguard personal information.

