# OpenReview forum: "Perturbation-Restrained Sequential Model Editing"
_ICLR.cc/2025/Conference — ICLR 2025 Poster_

### Official Review · Reviewer_KEXb · 2024-10-30

**Soundness:** 1
**Presentation:** 3
**Contribution:** 2
**Rating:** 5
**Confidence:** 5

**Summary:**

Knowledge editing methods when applied sequentially lead to significant model degradation. In this paper, the authors identify the reasons behind the loss of model's general abilities in the form of increasing singular value of edit-matrix as a function of the number of edits, and propose methods to control that.

**Strengths:**

1. The authors provide an interesting analysis of the potential reasons behind the loss of general ability of a model as it is edited sequentially.
2. The proposed methods is plug and play and can be applied on top of all existing parameter modifying model editing methods.

**Weaknesses:**

Many choices and decisions made in the paper seem unmotivated or not explained properly. I defer to the questions sections where clarifications are requested. I am happy to revise my review post the discussion period once the questions have been answered.

**Questions:**

1. In section 3.2, where the authors perform matrix perturbation analysis, it is not clear to me how that is relevant. The main objection I have is that if an MLP layer in a model is edited, everything until that layer remains the same. This means that "keys", which are by definition input to the edited matrix, remain identical before and after editing. Then I'm not sure what is the meaning of ∆k here, since the value of k never changes when an edit is made.

2. In line 241, the spectral norm definition is used, whereas line 203 says that ||*|| represents 2-norm.

3. The hypothetical situation described in line 271-275 is not clear to me. Does the situation involve making N sequential edits about the same fact? If that's the case, then each edit matrix will not be equal to ∆W_1 as the value of ∆W_1 depends on the base matrix, which would be been changed. Although I understand that this is not important in the larger scheme of things, I think it is not correct. I am happy to provide more explanation.

4. In Table 4, we can see that the singular value of the edited matrix increases way more dramatically for MEMIT compared to other methods. Why is that the case? Also it makes me question that the singular values are the only underlying cause for model degradation, as if that was the case, since MEMIT has the largest singular value blow up, it would be the least stable and see maximum model degradation. But that is not the case, in fact MEMIT is the most stable of the three. What is the author's take on this?

5. In equation 4 and 5, I would like some more intuition and explanation of the choice of restraining function. It is not obvious to me why the log function suggested by them works. I think this part of the paper has the least detail whereas I would have preferred a lot more detail here. For example, I would like to see more ablations on the different actions that can be performed on the larger singular values of the ∆W_N matrix.

6. From an intuitive understanding, this paper tries to attenuate the largest singular values of the edit matrix. But the largest singular values are the most important part of a matrix, providing the direction of largest variance and loosely speaking is the direction along which the most significant action is done by the matrix. How is attenuating the most significant action of a matrix not resulting in a lot of issues? Why is it okay to attenuate the largest singular values of a matrix? I think this is a much deeper question and warrants investigation.

7. The experiments shown in Figure 3 and 4 involve too few sequential edits. This method will be a lot more believable if the authors presented results for atleast 1000 edits, but preferably even more.

My final comment to the authors is that while this paper presents an interesting empirical contribution and figure 5 makes a very good empirical case for usefulness, I have a hard time understanding why the approach proposed by the authors works (questions 4-6). I do not find the explanations or motivations given in the paper appropriate so far. I am looking forward to hearing the author's comments during the discussion phase and am very happy to update my scores based on it if the authors are able to sufficiently clarify these questions.

---

> ### Comment · Reviewer_KEXb · 2024-11-20
> **Concluding remarks**
>
> I want to thank the authors for participating in the discussion phase. I have updated my scores.
>
> My final comments for the paper are:
>
> 1. While the condition number seems to be correlated with the loss of performance, I do not believe that the perturbation theory analysis, as presented by the authors, explains it. I believe this warrants further investigation.
>
> 2. A more in depth analysis of the relationship between condition number and loss of performance could have been explored in this paper by studying both these together. The authors haven't spend enough time on this analysis in the paper and I believe it is important in strengthening their claims.
>
> 3. The model editing performance with PRUNE is cut off too early in the paper to see a long term effects. As a reviewer, I cannot be certain of the impact of PRUNE unless I can see much longer term performances.
>
> These will be my final comments and updates.

---

### Official Review · Reviewer_1Sg3 · 2024-11-02

**Soundness:** 3
**Presentation:** 2
**Contribution:** 3
**Rating:** 6
**Confidence:** 4

**Summary:**

The research tackles the issues that arise due to weight shifts in sequential model editing when using weight-modifying approaches. The authors present a new approach, PRUNE, which aims to stabilize editing by maintaining control over these shifts. Importantly, the paper proposes an upper bound on weight modifications, beyond which the model's original capabilities degrade, resulting in reduced performance and an inability to retain edits. This bound serves as a theoretical safeguard, ensuring that model modifications remain within an optimal range for consistent, reliable performance.

**Strengths:**

The paper introduces a theoretically derived upper bound for weight modification. By defining this threshold, the authors provide a quantitative limit which when exceeded, triggers degradation in the model’s original functions and leads to edit forgetting. This contribution deepens understanding of the trade-offs involved in weight modification and offers a practical guideline for preserving model performance.

The authors demonstrate that PRUNE effectively mitigates the negative impacts of sequential edits. PRUNE shows considerable performance improvement by stabilizing the model against shifts in weight values, which may otherwise lead to inconsistencies or loss of prior knowledge.

**Weaknesses:**

The study focuses solely on the Llama-2 (7B) model, with a relatively small set of samples for evaluation which is a concern as editing in large numbers might exacerbate the impact of the shift that needs to be controlled. To broaden the scope, the authors could consider using a smaller model from the GPT series or a single editing approach, allowing for an expanded sample size under computational constraints. This could strengthen the study by providing more robust evidence for PRUNE’s effectiveness across different architectures. (major)

While PRUNE is effective, some edits may still be lost in the process, particularly when just applying the operation. A deeper analysis of this limitation could provide valuable insights. (minor)

**Questions:**

The PCA visualization discrepancy is not entirely clear maybe a slightly modified visualization would clarify what is being referred to.

The baseline capabilities of Llama on general abilities in Table 2 are missing.

---

### Official Review · Reviewer_FxE7 · 2024-11-04

**Soundness:** 3
**Presentation:** 3
**Contribution:** 3
**Rating:** 6
**Confidence:** 4

**Summary:**

This paper investigates the capability of large language models (LLMs) to preserve their general knowledge after sequential editing through a plug-and-play framework named PRUNE. The authors suggest that the condition number of the edited matrix significantly influences the general abilities of the edited models. They propose a method which involves SVD decomposition to ensure that the updated matrix maintains the singular values of the original matrix, thereby preventing substantial perturbations. The paper evaluates the effectiveness of this method across three popular editing methods and three large language models

**Strengths:**

1. Knowledge editing is an important topic, and addressing sequential editing is a challenging yet meaningful direction.
2. This paper is well motivated and the design of PRUNE clearly explained.
3. This paper proposes a plug-and-play method, which demonstrates significant improvements in sequential editing across various models.

**Weaknesses:**

1.  Some important details need further clarification. For example, in the editing setup, the key $k$ in the key-value pair $(k, v)$ typically does not change, and usually only $W$ and $v$ are modified. It is unclear why the paper focuses on analyzing the changes in $k_i$ instead of $v$ or $W$.
2.   The PRUNE method imposes constraints on parameters (as described in Equations (3), (4), and (5)), which might potentially reduce the model’s ability to retain other capabilities or preserve historical knowledge. However, the paper lacks sufficient analysis and investigation of the potential negative impacts on the efficacy of the most recent knowledge edits. A more detailed discussion in this regard would improve the overall quality of the analysis.
3.   A comparison with other sequential editing methods should be included in the related work section. Moreover, additional discussion is needed on works aiming at preserving the general capability of an edited model.
4.  The model requires utilizing historical $\Delta W$ in each editing operation, meaning that historical information must be reused. This could potentially lead to an increase in computational resources.

**Questions:**

See weaknesses.

---

### Official Review · Reviewer_oQtG · 2024-11-04

**Soundness:** 3
**Presentation:** 3
**Contribution:** 3
**Rating:** 8
**Confidence:** 2

**Summary:**

This paper analyzes how the condition number of the edited matrix affects the models' sensitivity to perturbations and proposes a framework called PRUNE to impose constraints that lower this sensitivity. Experiments demonstrate that PRUNE effectively preserves the general abilities of LLMs while maintaining strong editing performance across various tasks.

**Strengths:**

* Sequential editing is indeed a crucial research issue in knowledge editing, and this paper examines it from a relatively comprehensive perspective while proposing an excellent solution.

* The literature review is thorough.

* The evaluation scope is extensive, based on three representative LLMs, including GPT-2 XL, LLaMA-2, and LLaMA-3. It also includes four representative downstream tasks—reasoning, summarization, open-domain question answering, and natural language inference—to broadly demonstrate the impact of model editing on the general abilities of LLMs.

* The performance of the method proposed is good, effectively mitigating the decline in model general abilities caused by sequential edits across multiple metrics.

**Weaknesses:**

1. It would be helpful to report the results on the "Probability" metric, which is described in both [1] and [2]. I would like to know if applying the proposed framework to impose constraints on sequential editing affects the generalization and ripple effects of the edits themselves.

2. It would be more comprehensive if we could see some results on larger-scale models, such as at least those with 13B parameters or more.

3. Methods like MEMIT support batch editing, so it would be worthwhile to experiment on whether PRUNE can still maintain good model general ability in such batch editing scenarios.

---
**References**

[1] Evaluating the Ripple Effects of Knowledge Editing in Language Models

[2] Editing Large Language Models: Problems, Methods, and Opportunities

**Questions:**

Please see the Weaknesses section above.

---

### Meta-Review · Area_Chair_84KN · 2024-12-20

**Metareview:**

Summary of Scientific Claims and Findings

The paper's main claim is that the degradation of general model abilities during sequential model editing is closely linked to the condition number of the edited matrix. To address this, the authors introduce PRUNE, a framework that stabilizes the impact of edits by limiting the size of perturbations. Experiments across multiple LLMs (such as GPT-2 XL, LLaMA-2, and LLaMA-3) demonstrate that PRUNE is effective in preserving model performance across a variety of tasks. The authors show that applying PRUNE limits perturbations, reducing forgetfulness and improving stability even after multiple edits.

Key findings include:
1. PRUNE improves model performance by preventing excessive perturbations during sequential edits.
2. The framework is demonstrated to be effective across different LLM architectures and tasks, maintaining generalization ability and reducing memory loss.
3. Applying PRUNE multiple times improves model performance more than applying it once, highlighting the method’s scalability.

Strengths of the Paper

1. Theoretical Contribution: The paper offers a novel approach to mitigating performance degradation during sequential model editing by incorporating perturbation theory and proposing an upper bound for weight modification. This theoretical contribution provides a clear, actionable framework for stabilizing model performance.

2. Experimental Validation: The experiments conducted on LLaMA-2 (7B) and GPT-2 XL (1.5B) demonstrate the practical utility of PRUNE in real-world settings. The paper successfully shows that the method improves model stability and preserves general abilities in the face of long-term edits.

3. Clarity of Methodology: The paper clearly explains the rationale behind PRUNE, particularly the use of a logarithmic restraining function to control the growth of the largest singular values. This offers a concrete solution to the challenge of avoiding overfitting while retaining essential knowledge.

4. Practical Implications: The results suggest that PRUNE could be of significant value for practitioners who need to perform sequential edits on large models while minimizing the risk of knowledge loss.

Weaknesses and Missing Elements

1. Limited Scope of Models and Tasks: The experiments are primarily conducted on a small set of models (LLaMA-2 and GPT-2 XL) and tasks (reasoning, summarization, question answering). While these models are well-known, the results could benefit from broader evaluation on additional model architectures (such as GPT-3, T5, or BERT) and tasks, particularly those involving domain-specific or multi-modal learning. Expanding the sample size and task variety would enhance the generalizability of the results.

2. Impact of Multiple Edits on Recent Changes: Although PRUNE is shown to be effective for most sequential edits, the efficacy of the most recent edits diminishes after many applications of the method. Further exploration into this phenomenon and the design of more targeted techniques for recent edits could improve the approach.

3. Clarification on PCA Visualizations and Baseline Comparisons: There are mentions of discrepancies in PCA visualizations and baseline performance comparisons, particularly for LLaMA on general tasks. The authors should address these concerns in more detail to clarify the results and ensure full transparency.

4. Further Exploration of Method Limitations: The paper discusses the limitations of PRUNE, particularly in scenarios where edits may still be lost despite the constraints. Expanding this discussion and exploring alternative methods for mitigating this loss would strengthen the overall contribution. For instance, analyzing how PRUNE compares with other contemporary model-editing techniques like MEMIT or ROME could offer insights into its relative advantages and disadvantages.

Decision Rationale

The paper provides a novel and significant contribution to the field of model editing, particularly for large language models. The introduction of PRUNE is theoretically sound and experimentally validated, offering a solution to an important challenge in machine learning. The strength of the paper lies in its clear methodology, thorough experimental evaluation, and potential practical impact.

However, there are some limitations, primarily the narrow scope of models and tasks, the diminishing efficacy of PRUNE for recent edits, and the need for more thorough clarification in certain experimental aspects. Despite these weaknesses, the core contribution is strong, and the framework holds promise for further research and refinement.

Based on the strengths in theoretical innovation and practical application, and considering the possibility for further improvements and future research directions, I would recommend accepting this paper, with minor revisions to address the aforementioned weaknesses.

**Additional Comments On Reviewer Discussion:**

The authors' responses to the reviewer feedback were generally constructive and provided clarity on several points. They acknowledged the limitations of their current experiments, such as the limited model selection (mainly LLaMA-2 and GPT-2 XL) and the diminishing effectiveness of PRUNE for recent edits. The authors also noted that applying PRUNE multiple times rather than just once can improve stability, which is an important addition to the paper’s findings.

Overall, the authors have addressed the reviewers’ concerns with detailed explanations and additional experiments. While some issues remain, they are manageable and can be improved in revisions. The paper offers a valuable contribution to the field of model editing, and the suggested adjustments would enhance its clarity and applicability.

---

### Decision · Program_Chairs · 2025-01-22

Accept (Poster)